# FOXM1 Inhibition in Ovarian Cancer Tissue Cultures Affects Individual Treatment Susceptibility Ex Vivo

**DOI:** 10.3390/cancers13050956

**Published:** 2021-02-25

**Authors:** Luzie Brückner, Annika Reinshagen, Ngoc Anh Hoang, Anne Kathrin Höhn, Florian Lordick, Ingo Bechmann, Bahriye Aktas, Ivonne Nel, Sonja Kallendrusch

**Affiliations:** 1Institute of Anatomy, University of Leipzig, Liebigstr. 13, 04103 Leipzig, Germany; Luzie.Brueckner@medizin.uni-leipzig.de (L.B.); Ngoc.Hoang@medizin.uni-leipzig.de (N.A.H.); Ingo.Bechmann@medizin.uni-leipzig.de (I.B.); 2Department of Gynecology, University of Leipzig Medical Center, Liebigstr. 20a, 04103 Leipzig, Germany; Annika.Reinshagen@medizin.uni-leipzig.de (A.R.); Bahriye.Aktas@medizin.uni-leipzig.de (B.A.); Ivonne.Nel@medizin.uni-leipzig.de (I.N.); 3University Cancer Center Leipzig (UCCL), University of Leipzig Medical Center, Liebigstr. 22, 04103 Leipzig, Germany; Florian.Lordick@medizin.uni-leipzig.de; 4Institute of Pathology, University of Leipzig Medical Center, Liebigstr. 26, 04103 Leipzig, Germany; AnneKathrin.Hoehn@medizin.uni-leipzig.de; 5Department of Oncology, University of Leipzig Medical Center, Liebigstr. 22, 04103 Leipzig, Germany

**Keywords:** ovarian cancer, FOXM1, tissue culture, resistance, homologous recombination (HR), susceptibility, olaparib, carboplatin, thiostrepton

## Abstract

**Simple Summary:**

Late diagnosis of ovarian cancer is a major reason for the high mortality rate of this tumor entity. The time to determine tumor susceptibility to treatment is scarce and resistance to therapy occurs very frequently. Here, we aim for a model system that can determine tumor response to (I) study novel drugs and (II) enhance patient stratification. Tissue specimens (*n* = 10) were acquired from fresh surgical samples. Tissue cultures were cultivated and treated with clinically relevant therapeutics and an FOXM1 inhibitor for 3–6 days. The transcription factor FOXM1 is a key regulator of tumor survival affecting multiple cancerogenic target genes. Gene expression of FOXM1 and its targets BRCA1/2 and RAD51 were investigated together with tumor susceptibility. Tissue cultures successfully demonstrated the individual benefit of FOXM1 inhibition and revealed the potency of the complex model system for oncological research.

**Abstract:**

Diagnosis in an advanced state is a major hallmark of ovarian cancer and recurrence after first line treatment is common. With upcoming novel therapies, tumor markers that support patient stratification are urgently needed to prevent ineffective therapy. Therefore, the transcription factor FOXM1 is a promising target in ovarian cancer as it is frequently overexpressed and associated with poor prognosis. In this study, fresh tissue specimens of 10 ovarian cancers were collected to investigate tissue cultures in their ability to predict individual treatment susceptibility and to identify the benefit of FOXM1 inhibition. FOXM1 inhibition was induced by thiostrepton (3 µM). Carboplatin (0.2, 2 and 20 µM) and olaparib (10 µM) were applied and tumor susceptibility was analyzed by tumor cell proliferation and apoptosis in immunofluorescence microscopy. Resistance mechanisms were investigated by determining the gene expression of FOXM1 and its targets BRCA1/2 and RAD51. Ovarian cancer tissue was successfully maintained for up to 14 days ex vivo, preserving morphological characteristics of the native specimen. Thiostrepton downregulated FOXM1 expression in tissue culture. Individual responses were observed after combined treatment with carboplatin or olaparib. Thus, we successfully implemented a complex tissue culture model to ovarian cancer and showed potential benefit of combined FOXM1 inhibition.

## 1. Introduction

Resistance or acquired resistance to therapy is one of the main causes of high mortality rates in oncology. Patients with ovarian cancer show the highest mortality among all gynaecologic cancers and are mostly diagnosed at an advanced stage which requires complex therapy and is accountable for an average five-year survival rate of 46.5% [1,2]. Surgical resection combined with chemotherapy is the standard of care, as indicated by guidelines for ovarian cancer [2,3]. First line treatment after an optimal surgery without residual tumor is a combination of carboplatin and paclitaxel which initially shows good response rates [4,5]. However, almost one third of the patients are non-responders and recurrence appears in 70–80% of all patients, significantly affecting survival [6,7,8]. Second-line therapy is usually not curative and often shows resistance towards platinum which may be acquired during initial treatment [7,8]. Still, only few patients are eligible for targeted therapy. In current clinical practice, maintenance therapy with PARP inhibitors after chemotherapy is considered for patients with BRCA mutations. Also, the additional administration of bevacizumab targeting angiogenesis was shown to be effective [2]. Besides, more targeted therapeutics are currently under investigation in numerous clinical and preclinical trials [9,10,11]. Among them, thiostrepton, a thiopeptide antibiotic used for bacterial infections in veterinary medicine, is a promising approach in the treatment of different cancer entities [12,13,14,15]. In ovarian carcinoma first beneficial evidence for thiostrepton administration exists in cell culture and murine models reducing tumor volume in vivo [16,17,18]. This effect seems to be based on the inhibition of FOXM1, which is proposed to be achieved through proteasome inhibition by stabilizing a negative regulator of FOXM1 and direct interaction with FOXM1 [19,20]. FOXM1 is a transcription factor which is a key regulator of different oncogenic signaling pathways and is frequently upregulated in ovarian cancer [21,22,23]. Expression of FOXM1 in malignant tumors was shown to associate with high-grade disease, therapeutic resistance and poor prognosis [23,24,25]. Through direct and indirect downstream regulation of a broad spectrum of genes, FOXM1 plays an important role in proliferation, cell cycle control, DNA repair and thus in tumorigenesis, cancer progression and tumor growth [26,27]. In its function as a transcription factor, FOXM1 appears to alter the expression of target genes involved in homologous recombination (HR) in DNA repair such as BRCA1/2 and RAD51 [28,29,30,31]. In this context, chemotherapeutics targeting DNA are known to trigger DNA damage repair mechanisms (e.g., HR in tumor cells) which may represent a mechanism of resistance [32,33,34]. Also, FOXM1 is suspected to be upregulated by olaparib which inversely correlates with sensitivity to the PARP inhibitor [35]. Hence, the downregulation of those DNA repair genes through FOXM1 inhibition by thiostrepton could be beneficial in combination with olaparib or carboplatin resulting in DNA damage mediated cell death and overcoming resistance [18,31,35,36]. The aim of the current study was to investigate the effects of thiostrepton in a more complex setting using tumor tissue cultures from patient-derived specimens. This model, established in other solid tumor entities such as gastric, lung and colorectal cancer, is capable of use to study tissue markers including FOXM1 and allows the investigation of tumor cells in their microenvironment (TME) which could play an important role in assessing drug response and interactions [37,38,39,40]. Comprehension of tumor cell interaction within its surrounding is key to understand the underlying mechanisms of tumor evolution and resistance development. Thereby, tissue culture depicts a suitable model for preclinical susceptibility testing in individual patients and enhances comparability with tumor conditions found in vivo [41]. Here we present a stable and reproducible method that enables the investigation of individual tissue responses to optimize future therapeutic options and effective individual treatment for ovarian carcinoma patients.

## 2. Results

### 2.1. Ovarian Cancer Tissue Ex Vivo

Ovarian cancer specimens were successfully cultivated up to 14 days on an air-liquid interface model (Figure 1, Appendix A). Slice cultures showed a good preservation of morphologic features of the original tumor (Figure 1A). Parameters including tumor cell formation, serous-papillary configuration and typical stroma presentation were assessed for comparison from 14 tumor specimens. All tumor samples were characterized as serous adenocarcinomas by the institute of pathology. Tumor localization was specified as ovarian or tubal origin which is referred to in the text as “ovarian cancer”. In all samples, the pathological diagnosis of the original tumor matched the features found in baseline culture, representatively shown in Table 1. 

Some cases (4 of 14) were excluded from the analysis due to the lack of malignant tumor tissue in the obtained samples, technical difficulties or insufficient preservation. Specimens not shown in the table were used for adjustments of the culture conditions. Figure 1 demonstrates the maintenance of morphological characteristics of one representative specimen up to day 14 ex vivo. The tumor cells as well as the tumor microenvironment are well-preserved. Immunological cell subtypes (cytotoxic/t- lymphocytes and macrophages) were stained and microscopically detected. All cell types could be identified up to day 14 in culture. The overall tumor fraction of 60 ± 5,4% (mean ± SEM) in the baseline control was preserved up to day 14 in culture (day 4: 57,5 ± 1,5%; day 7: 46 ± 5,7%; day 14: 53 ± 15%) (Figure 1B). However, tissue cultures adapted to the culture conditions and showed altered proliferation. This adaption is critical; therefore, experiments were conducted within four days. Within this time boundary, tumor tissue cultures of ovarian cancer enable drug testing as well as the investigation of the complex interplay between tumor cells and its organotypic surrounding. 

### 2.2. Effects of Cytotoxic Drugs on Tumor Cell Proliferation and Apoptosis 

Tissue cultures were treated with different doses of carboplatin (0.2 µM, 2 µM and 20 µM). Tumor cell proliferation showed a decrease at 0.2 µM of carboplatin but remained stable at the higher doses (2, 20 µM) of carboplatin compared to the untreated tissue samples. The apoptotic tumor cell rate increased dose-dependently (Figure 2). Individual cases displayed the broad variance of individual tissue response. Case #10 seen in Figure 2 shows a histological response characterized by tissue reorganization and decrease of the tumor fraction after treatment with carboplatin. Tumor proliferation decreased at all applied doses of carboplatin. Also, there was an increase of apoptotic tumor cells observable at all carboplatin conditions. Another tissue specimen (#11) hardly responded to carboplatin supplementation considering tumor proliferation, whereas a dose-dependent increase of the apoptotic tumor fraction was seen. Within the cultivated tissue, various areas can be discriminated. Included in the analysis, regions with lower oxygen and nutritional supply were detected localized in the slice center. Besides these regions, interestingly, adjacent cells showed strong proliferation. The observed selection of tumor cells with the most beneficial characteristics causing strong proliferation could play a major role inducing recurrence after initial treatment. In summary, tissue specimens that do not respond to carboplatin treatment can be selected in the ex vivo model and response patterns can be assessed leading to the declaration of response, partial response or non-response (Table 1).

### 2.3. Response to the FOXM1 Inhibitor Thiostrepton

To investigate the potency of thiostrepton to downregulate FOXM1 in this heterogeneous setting, we applied a clinically relevant dose of 3 µM and investigated tissue response as well as the molecular tissue adaptation. Thiostrepton treatment resulted in a modest inhibition of tumor cell proliferation and partially enhanced apoptotic rates of the tumor cells (Figure 3). Moreover, thiostrepton downregulated FOXM1 and all investigated target genes of HR (BRCA1/2, RAD51) on mRNA level compared to the vehicle control (Figure 4A, B, *n* = 3). One specimen (#20) massively upregulated BRCA2 and RAD51 expression after thiostrepton supplementation (Figure 4C). Nevertheless, the proliferation rate of that case was decreased and apoptosis enhanced by monotherapy with the FOXM1 inhibitor, indicating diverse regulation of FOXM1 signaling (Appendix A). Regarding the detectable but moderate effect on ovarian tissue cultures by thiostrepton alone, a possible benefit of combining it to the PARP inhibitor olaparib and the current standard of care agent carboplatin was investigated in the established model.

### 2.4. Combined Effect of Thiostrepton and Olaparib

The PARP inhibitor olaparib is clinically approved for patients carrying a BRCA mutation, inducing synthetic lethality of tumor cells by disrupting the compensatory non-homologous end-joining. The combined treatment with thiostrepton and olaparib provokes downregulation of BRCA1/2 through inhibition of FOXM1, referred to as “BRCAness” in tumors without BRCA mutation by Fang et al. (2018) [35]. Here, we administered thiostrepton together with olaparib to counter the alteration of HR which is suspected to be one responsible effect for resistance to PARP-inhibition.

Overall analysis (*n* = 7) did not show additive effects on tumor proliferation rates in this combination (Figure 3A). However, tumor apoptosis increased compared to olaparib monotherapy (Figure 3B). Individual cases revealed great diversity though (Figure 5, Appendix A). 

Case #17 (Figure 5) shows a significant inhibition of proliferation to about 48 ± 4.3% (mean ± SEM) with olaparib as monotherapy compared to the vehicle control. The same effect was observed with the combination of thiostrepton and olaparib (44.7 ± 9.5%). In contrast, apoptotic tumor cell rate heterogeneously altered to about 708 ± 507% after combined treatment application, whereas olaparib alone enhanced the apoptotic tumor cell rate to only 210 ± 100%. This case also showed strong downregulation of FOXM1, BRCA1/2 and RAD51 mRNA transcripts by thiostrepton, and together with olaparib, while olaparib alone triggered the already prescribed upregulation of all considered mRNA transcripts. The mentioned case confirms the gene regulation induced by thiostrepton and olaparib described in literature based on cell line experiments (Figure 4A and Figure 5). 

In contrast to this finding, the combination of thiostrepton and olaparib did not have significant benefits in #22. Tumor cell proliferation and apoptosis could not be regulated in all triplets, indicating resistance of some tumor populations. While FOXM1, BRCA1/2 and RAD51 transcripts were downregulated by thiostrepton, mRNA expression of BRCA2 and RAD51 remained stable or increased after olaparib and combined application with thiostrepton, respectively (Figure 5). The depicted results suggest that a part of ovarian cancer patients could benefit from the additive administration of thiostrepton to olaparib. Expression recognition is therefore a helpful tool to evaluate tissue response and underlying mechanisms in the shown cases. 

### 2.5. Combined Effect of Thiostrepton and Carboplatin

As FOXM1 expression is frequently upregulated in ovarian cancer tissue and is suggested to support chemoresistance [22,23,36], we investigated whether an additional benefit can be achieved by combination of the first-line agent carboplatin with thiostrepton. Platinum-based chemotherapy was also shown to enhance HR as a DNA damage repair reaction, potentially upregulating BRCA1/2 and RAD51, so that an additional effect on tumor cell death of thiostrepton can be assumed [23]. Taken all specimens together, carboplatin (2 µM) showed no effect on tumor cell proliferation, however the apoptotic tumor cell rate was non-significantly increased. The additional supplementation of thiostrepton moderately decreased the tumor proliferation and enhanced the apoptotic tumor cell rate in a greater subpopulation of specimens (Figure 3). Examining each individual specimen, we could again confirm the patient specific adaptation observed in gene expressions of FOXM1 and downstream HR (Fig 5). In specimen #17 a benefit was seen supplementing carboplatin and thiostrepton together which resulted in reduced tumor cell proliferation and enhanced apoptotic tumor cell rates compared to single carboplatin administration (Figure 5). However, #17 was not susceptible to single carboplatin or thiostrepton treatment. The combined effect could be confirmed by RT-qPCR showing that the mRNA expression of BRCA2 and RAD51 was upregulated with carboplatin while thiostrepton addition encountered this expression pattern effectively (Figure 5). Specimen #22 reacted with altered apoptotic tumor cell rates on carboplatin supplementation. Tumor cell proliferation, however, also increased, indicating the development of resistance. Additional supplementation of thiostrepton decreased both, the tumor cell proliferation, and the apoptotic tumor cell rate. In #22 gene expression analysis revealed possible compensatory upregulation of BRCA2 and RAD51 indicating low susceptibility to this treatment regimen (Figure 5). However, the tumor tissue culture model of ovarian carcinoma appears to be suitable for stratification of patients that might benefit from clinically applicable therapeutics plus thiostrepton and pathways involved in the FOXM1 regulation might be comprehended.

## 3. Discussion

Models offering a robust platform for accurate studies of interaction between tumor cells and their microenvironment are crucial for the development of novel therapeutics. Here, we present a highly standardized tissue culture model derived from ovarian carcinoma. Tissue slice cultures were proven to maintain morphologic characteristics featuring the tumor microenvironment and depict metabolic challenges which induce hypoxic areas and limited nutrition supply, mimicking in vivo tumor conditions [37,40,42,43]. Preservation of morphology as well as the composition of the original tumor tissue in our cultures confirm once more the current approaches. Featuring complete oncological hallmarks, clinical correlation could meanwhile also be demonstrated in different tumor entities and ovarian cancer by others [41,44,45]. Therefore, tumor slice cultures can successfully be used for developing individualized treatment attempts and study cancer biology [39,46,47]. Confirming that our model reflects treatment response ex vivo, an overall dose-dependent effect of carboplatin, the established first line treatment in ovarian cancer, on apoptosis, was observed. However, each specimen displayed individual susceptibility, verifying the heterogeneity of patient-derived material which is complex and thereby representative for the original tumor.

Here, the tissue model was used to investigate the potency of FOXM1 inhibition in ovarian cancer specimens, as the transcription factor network is altered in 87% of ovarian cancer patients [48]. Overexpression of FOXM1 in high-grade ovarian cancer patients was shown to correlate with resistance against platinum-agents and poor prognosis, with patients quickly building up resistance to standard therapy which was also observed with ovarian cancer cell lines in vitro [23,49,50,51]. Thiostrepton treatment downregulated FOXM1 expression on mRNA level in ovarian tissue cultures, confirming results of cell culture experiments [16,17]. Investigating tumor survival by proliferation and apoptosis of tumor cells, tissue cultures displayed a more differentiated picture than in common tumor models. This discrepancy might be attributable to the innate differences of the immanent tissue properties, containing different cell types and stromal substance which might influence gene expression.

Additionally to single thiostrepton administration, FOXM1 inhibition could improve the effect of olaparib and carboplatin in cell culture and xenograft models [18,35,36]. Therefore, another focus of the present study was to investigate the beneficial effect of FOXM1 inhibition in addition to standard therapy in the tissue culture model. Findings from cell and mouse models were reproducible in the current study for individual cases investigating tumor cell survival and show that thiostrepton improves treatment response compared to single agent effects. The pathway by which the described benefit could be explained was further evaluated. As described in preliminary literature, platinum derivates as well as PARP inhibitors might induce the activation of HR, provoking tumor resistance [30,35]. FOXM1 inhibition was therefore shown to counteract this tumor adaption of increased HR and to downregulate BRCA1/2 and RAD51 [30,35]. Our results reveal that the PARP inhibitor olaparib increased BRCA1/2, RAD51 and FOXM1 expression, confirming previously described findings [35]. However, individual regulation varied in the different cases. Carboplatin, otherwise, did not upregulate these targets in all of the shown cases assuming other pathways of resistance development [52,53]. Supporting the mentioned pathway mechanism, FOXM1 inhibition by thiostrepton could successfully overcome downstream BRCA1/2 and RAD51 expression administered together with olaparib in cases responsive to combined treatment. 

However, one case showed massive upregulation of the HR target genes on a transcriptional level after thiostrepton single treatment, while showing downregulation after combined regimen. Nevertheless, tumor response, seen in apoptosis and proliferation rate, is detected after monotherapy as well as the combination. Unexpected ways of response could be explained by the fact that FOXM1 regulates various downstream target genes and BRCA2 is also affected by different regulators [26,31], suggesting other involved pathways which are currently under investigation in cell culture studies [54,55]. Thiostrepton further is a potent proteasome inhibitor and inhibits the degradation of pro-apoptotic factors, indicating another cause of apoptotic effects on tumor cells [56,57]. Not completely clarified pathways of thiostrepton involve the immunological compartment and the possible function as an enhancer of immune-mediated effects together with chemotherapeutics. Therefore, a reduced number of regulating T-cells (Tregs) in the TME and reduced tumor size in immunocompetent mice were shown when combining thiostrepton to oxaliplatin [58]. Although the connection of FOXM1 to the TME is not yet well understood, it is suggested to play a role in T-cell differentiation and to affect the proliferation of macrophages [59,60]. Considering an important influence of immune cells in cancer development and treatment, tissue cultures display a vital immunological compartment [40,61], and thus might help to understand the complex FOXM1 network in context of the TME and the effects of thiostrepton by modulating immune activation. While our findings have set the base for investigations on ovarian carcinoma tissue culture, further work should also take clinical patient data into consideration for correlative comparison and evaluation with ex vivo results.

## 4. Materials and Methods 

### 4.1. Specimens

Tumor specimens were obtained from patients treated at the University of Leipzig Medical Center, Germany. A total of 14 ovarian cancer patients were included in this study, which was approved by the Ethics Committee of the Medical Faculty, University of Leipzig (216/18-ek). All patients provided their informed written consent to this study.

### 4.2. Ovarian Cancer Tissue Preparation

The culture protocol was previously described and applied with modifications [39]. Briefly, after surgical resection and macroscopic assessment by a pathologist, tumor samples were cut into slices of 350 μm using a tissue chopper (McIlwain TC752; Campden Instruments, Lafayette, MA, USA) (Appendix A). Tissue slice diameter was standardized using a clinical 3 mm skin punch (kai Europe, Solingen, Germany). Three tissues were placed together onto one membrane insert (Merck Millipore, Billerica, MA, USA) and cultivated in a six-well plate. One insert formed one condition of one tumor specimen. A controlled randomization needs to be conducted designing the tissue triplets considering tumor heterogeneity. Tissues were incubated under standardized conditions of 37 °C and 5% CO_2_. Medium (phenol-free RPMI 1640 (Thermo Fisher Scientific, Waltham, MA, USA)), supplemented with 1% penicillin/streptomycin (Merck, Darmstadt, Germany; 10,000 U penicillin/10 mg/mL streptomycin in 0.9% NaCl), 1% L-glutamine (Thermo Fisher Scientific, 200 mM) and 10% fetal calf serum (Thermo Fisher Scientific) was changed 2–3 h from preparation and every other day after preparation unless stated otherwise. Tissues were fixated overnight using 4% paraformaldehyde (Thermo Fisher Scientific) at the day of preparation (baseline) and after cultivation. In experiments using olaparib and thiostrepton, DMSO controls were taken. Baseline controls of day 0 served as reference of tissue maintenance.

### 4.3. Tissue Maintenance and Drug Treatment

On day 2, 4, 7 and 14 ex vivo, fixation of the tissue was performed using 4% paraformaldehyde (Thermo Fisher Scientific). Optimized culture medium (RPMI, 10% FBS, 1% L-glutamine) was replaced after 2–3 h from preparation. When available, 8% FBS and 2% of patient’s serum was supplemented (*n* = 5). A total of 24 h after tumor resection, different drugs were applied. Tissue cultures from 10 specimens were treated with carboplatin (0.2, 2 and 20 μM, *n* = 2 or 2 µM, *n* = 7), olaparib (10 μM) and thiostrepton (3 μM) as monotherapies and in combination (*n* = 7). Untreated inserts and DMSO equivalents were taken in each experiment and served as controls. All specimens on which drug susceptibility was tested did not receive neoadjuvant chemotherapy in vivo.

### 4.4. Staining Procedure and Analysis

The fixated tissue was embedded in paraffin and supplied so serial cutting. Slices of 5 μm were brought up on microscope slides (Agilent, Santa Clara, CA, USA). Hematoxylin and eosin (H&E) staining was performed for evaluation of morphology and tissue heterogeneity. For immunofluorescence staining, paraffin slices were deparaffinized. After heat-mediated antigen retrieval, the sections were washed with 0.3% Triton X in PBS and blocked with Normal Goat Serum (Jackson ImmunoResearch Laboratories, Inc., West Grove, PA, USA) for 30 min. Slices were incubated with primary antibodies diluted in 0.5% BSA at 4 °C overnight. After washing the slides with 0.3% PBS/Triton X, labeling with secondary antibodies was performed for 1 h. Nuclei were stained with Hoechst 33,342 (Sigma-Aldrich, St. Louis, MO, USA). For immunohistochemical staining with DAB, slices were deparaffinized and heat-mediated antigen retrieval was performed. The endogenous peroxidase was blocked with 0.5% H_2_O_2_ (Carl Roth, Karlsruhe, Germany) in PBS for 10 min and the sections were blocked with Normal Goat Serum (Jackson ImmunoResearch Laboratories, Inc.) for 30 min. Slices were incubated with Extra Avidin Peroxidase (Sigma Aldrich) for 60 min, rinsed with TRIS-buffer and color reaction was developed with DAB-tablets (Sigma Aldrich). Counterstaining was done with hemalaun (Hollborn & Söhne GmbH, Leipzig, Germany). All immunological markers (CD3, CD8, CD68) were stained as described here. For DAB staining of FOXM1, the Dako Real EnVision Detection System Peroxidase/DAB+, Rabbit/Mouse (Agilent) was used according to the manufacturers protocol. Head-mediated antigen retrieval was performed, and nuclei were stained with hemalaun (Hollborn & Söhne GmbH).

### 4.5. Reagents and Antibodies

Carboplatin and olaparib were purchased from Selleck Chemicals (Houston, TX, USA). Thiostrepton was obtained from Sigma-Aldrich. Pan-cytokeratin antibody was acquired from Bio Genex Laboratories (San Ramon, CA, USA), Ki67-antibody from DCS (Hamburg, Germany), cleaved-PARP-antibody from Abcam (Cambridge, UK) and FOXM1-antibody from Cell Signaling Technology (Danvers, MA, USA). Antibody against CD3 was purchased from Bio-Rad Laboratories, Inc. (Hercules, CA, USA), antibody against CD8 from Cell Signaling Technology and antibody against CD68 from Agilent. Secondary antibodies for immunofluorescence Alexa Fluor 568 goat anti-mouse antibody and Alexa Fluor 647 goat anti-rabbit antibody were obtained from Invitrogen (Carlsbad, CA, USA). Secondary biotinylated goat anti-rat antibody was purchased from Vector Laboratories (Burlingame, CA, USA) and goat anti-mouse antibody was acquired from Sigma Aldrich for immunohistochemical analysis.

### 4.6. Real-Time Quantitative PCR 

Total RNA was extracted from ovarian carcinoma tissue using TRIzol reagent (Invitrogen) according to the manufacturer’s protocol. After extraction, RNA was treated with TURBO DNA-free Kit (Invitrogen). The ProtoScript First Strand cDNA Synthesis Kit (New England BioLabs, Ipswich, MA, USA) was used for cDNA synthesis, with 1 μg of RNA in a 20 μL reaction. The resulting sample was diluted (1:2.5) with nuclease-free water and stored at −20 °C until further use. A total of 2 μL of cDNA were used with Maxima SYBR Green qPCR Master Mix (Thermo Fisher Scientific) on a CFX96 Real-Time PCR Detection System (Bio-Rad) including a non-template and a no-primer negative control for each assay. Tested samples were carried out in duplicates. RPLP0 and PPIA were used as reference genes and their amplification served for normalization of the mRNA levels. Sequences of the primer pairs are listed in Appendix A.

### 4.7. Quantitative Analysis of Immunofluorescence Markers

Stained microscope slides were scanned with the Leica Aperio Versa 8 Pathology Scanner (Leica, Wetzlar, Germany). Subsequently, the analysis was performed with the Cellular IF Algorithm in Aperio Image Scope (Leica). Parameters affecting cell size, staining localization and staining intensity/threshold were set manually and the software detected cell amounts automatically. Markups of the analyzed images were created by the software and controlled for adequacy. The settings were adjusted for single slides, when the markup did not match with the visual evaluation due to differences in tissue morphology, staining aspects and exposure times at scanning. Three classes of cells were determined: 1. cells, which were positive for pan-cytokeratin (tumor cells), 2. Ki67 (proliferating cells) or cleaved-PARP (apoptotic cells), respectively, and 3. cells with colocalization of both markers (panCK/Ki67 or panCK/cPARP). Apoptotic and proliferating cells were stained in horizontal cuts of neighboring regions. The amount of colocalized cells, which is referred to as proliferating or apoptotic tumor cells, was set in proportion to overall tumor cells. For the tumor cell fraction, tumor cells were set in proportion to all cells (counter-stained with Hoechst). Means of controls (CTR = culture medium or DMSO control) were set to 100% and treatment conditions were compared proportionately. Overview of this method is found in Appendix A.

### 4.8. Statistical Analysis

The total cell count (Hoechst positive), tumor cell count (Hoechst and cytokeratin positive) and proliferating or apoptotic tumor cell count (Hoechst, cytokeratin and Ki67/cPARP positive) was acquired for every whole slice scan. Mean slice values from the triplicates were then calculated to obtain the mean value for each condition. Slices not containing tumor or showing large necrotic areas were excluded from the triple set. Students *t*-test and One-way-ANOVA with Bonferroni correction was performed using GraphPad Prism 9 (GraphPad Software, La Jolla, CA, USA). *p* < 0.05 was considered significant. Significances were not tested for single RT-qPCR experiments (*n* = 1).

## 5. Conclusions

In conclusion, our study provides two key propositions. First, comprehensive understanding of FOXM1 regulation will provide merit for cancer therapy. Here we show that FOXM1 regulation by thiostrepton decreases tumor cell survival in addition to carboplatin or olaparib treatment in individual cases. Determining molecular responses, thiostrepton was shown to overcome suspected resistance induced by HR. Secondly, we provide the rationale for considering tissue culture as an important model to investigate tumor markers possibly influencing resistance to clinically applicable therapies. Further work might make this technique useful for personalized clinical stratification and enhanced response rates.

## Figures and Tables

**Figure 1 cancers-13-00956-f001:**
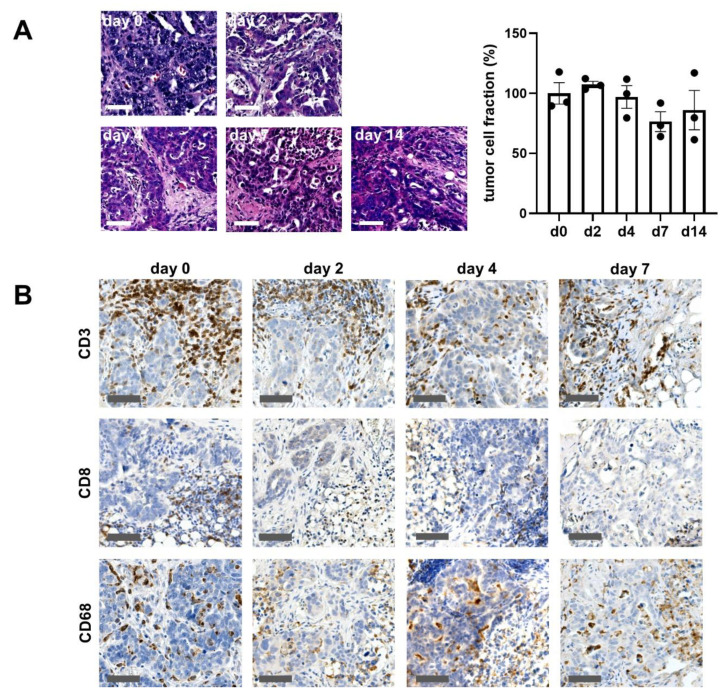
Tissue preservation ex vivo. (**A**) Tissue culture maintains its architecture and cellular composition throughout the cultivation period of 14 days. Tissue was fixated at the indicated days, embedded in paraffin and stained with H&E. Here, tissue was cut horizontally, 5 µm thick. Bar = 100 µM. The tumor fraction was stable up to day 14. Each dot represents one specimen (*n* = 3), d = day. Day 0 was set to 100%. Error bars show SEM (**B**) Tissues were stained with antibodies against CD3, CD8 and CD68 on day 2, 4 and 7 and were compared to day 0. Bar = 100 μm. (*n* = 3). One-way ANOVA (*p* < 0.05) showed no significant alterations.

**Figure 2 cancers-13-00956-f002:**
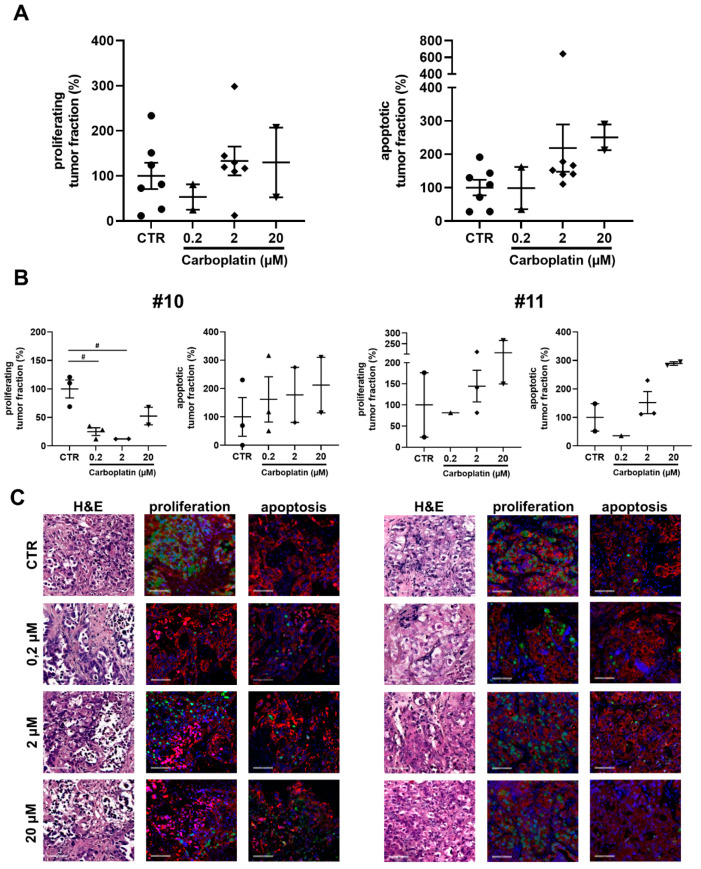
Dose-dependent responses with carboplatin treatment. (**A**) Effect of carboplatin on tumor proliferation and apoptosis. Tissue was treated with 0.2, 2 or 20 µM of carboplatin for 72 h after 24 h from the start of cultivation (*n* = 7 for CTR and 2 µM, *n* = 2 for 0.2 and 20 µM). Error bars represent SEM. (**B**) Individual response of #10 and #11 to carboplatin determined by tumor proliferation and apoptosis. Error bars represent SEM. (**C**) Representative images of #10 and #11. Tissue was stained with antibodies against Ki67 and cleaved-PARP. Bar = 100 μm. Blue = Hoechst, red = panCK, green = Ki67 in proliferation/cPARP in apoptosis. # *p* < 0.01 (One-way ANOVA).

**Figure 3 cancers-13-00956-f003:**
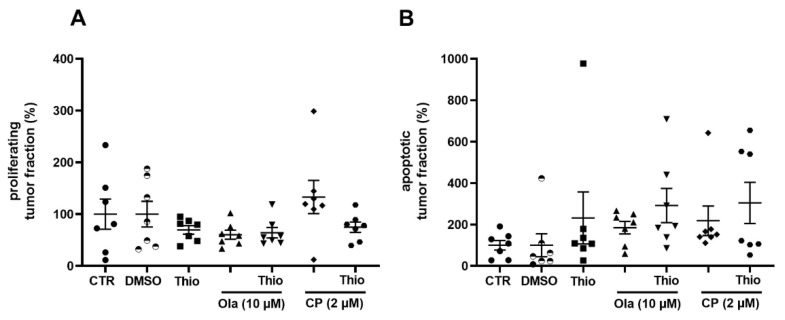
FOXM1- and PARP- inhibitor and carboplatin as mono- and combined therapy. Tissue was treated with thiostrepton (Thio), olaparib (Ola) and carboplatin (CP). To investigate the beneficial effect of FOXM1 inhibition, olaparib and carboplatin were combined with thiostrepton. DMSO is the vehicle control for all conditions except carboplatin. Conditions were normalized to their control, respectively. All drugs were applied for 72 h after 24 h from the start of cultivation. (**A**) Effect of all treatment conditions on tumor proliferation. Tissue was stained for Ki67, panCK and Hoechst to determine tumor proliferation. *n* = 7 (**B**) Effect of all treatment conditions on tumor apoptosis. Tissue was stained for cleaved PARP, panCK and Hoechst to determine tumor apoptosis. *n* = 7; error bars represent SEM. One-way ANOVA (*p* < 0.05) showed no significant alterations.

**Figure 4 cancers-13-00956-f004:**
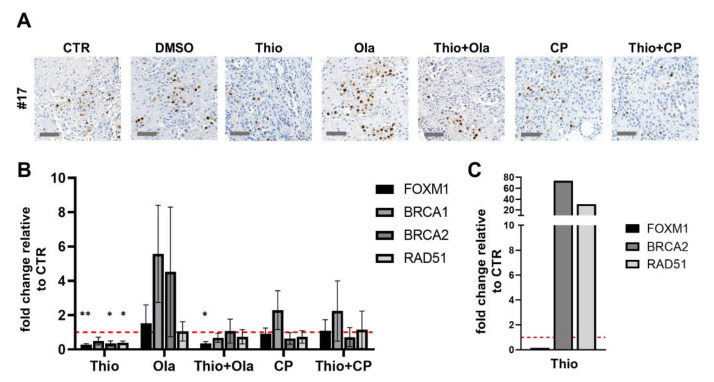
Regulation of the transcription factor FOXM1 and its downstream targets by application of thiostrepton (Thio), olaparib (Ola), and carboplatin (CP). (**A**) Tissue cultures were stained with FOXM1-antibody, representative pictures are shown. (**B**) Thiostrepton successfully downregulated all investigated gene expressions (FOXM1, BRCA1/2, RAD51). Olaparib, however, upregulated FOXM1 and all downstream targets while the addition of thiostrepton encountered this reaction. Carboplatin treatment only showed increased BRCA1 and thiostrepton could not overcome this effect in overall analysis. *n* = 3, error bars represent SEM, vehicle CTR was set to 1. (**C**) #20 expressed counter-regulation by BRCA2 and RAD51 investigating the FOXM1 inhibitor thiostrepton while FOXM1 was effectively downregulated. Vehicle CTR was set to 1. * *p* < 0.05, ** *p* < 0.005 (student’s *t*-test). Significance was not tested for *n* = 1.

**Figure 5 cancers-13-00956-f005:**
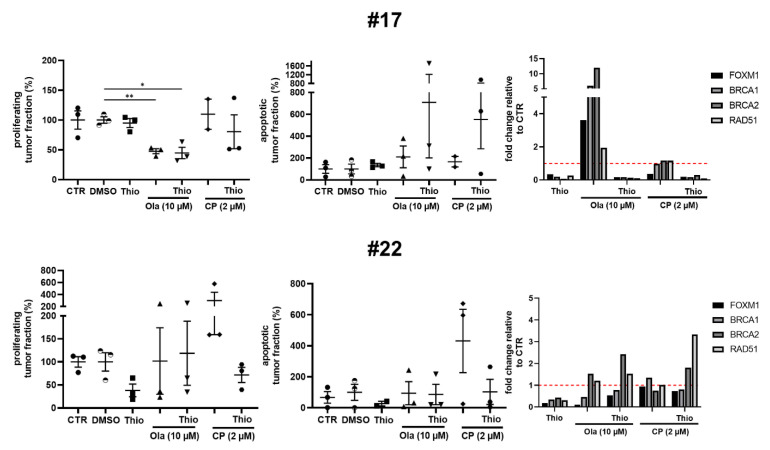
Regulation of the transcription factor FOXM1 and its downstream targets by application of thiostrepton, olaparib and carboplatin. Case #17 showed no beneficial effect of combination treatment (Thio + Ola/Thio + CP) in proliferation rates. Apoptosis was heterogeneously increased in the slice triplicates, showing an additive effect of thiostrepton. While olaparib strongly upregulated FOXM1, BRCA1/2 and RAD51, thiostrepton addition could overcome compensatory homologous recombination (HR) gene regulation. Similar findings could be seen for carboplatin (CP). #22 neither showed treatment response to Thio + Ola/Thio + CP in the quantification of proliferation or apoptosis, nor in mRNA expression. Error bars represent SEM. * *p* < 0.05, ** *p* < 0.005 (student’s *t*-test).

**Table 1 cancers-13-00956-t001:** Patient data and experimental tissue response.

#	Age	TNM (2017)	Grade	BRCA1/2	PS	ICD-O-C	Days in Culture	Treatment Period	Tissue Response	
									CP	Ola
1	83	pT3c pN1a M0	G3	n.d.	-	57	≤14	-	-	-
3	62	pT3c pN0 M0	G3	n.d.	-	57	≤14	-	-	-
6	60	pT3a pN1a M0	G3	BRCA2_mut_	-	57	≤14	-	-	-
10	79	pT3c pN1b M0	G3	n.d.	+	57	≤7	72 h/6 d	R	R
11	67	pT3c pN1b M0	G3	no mut.	-	57	≤7	72 h/6 d	PR	PR
12	64	pT3c pN1b M0	G3	no mut.	-	57	4	72 h	NR	R
17	63	ypT3c ypN0 M0	G3	no mut.	+	56	4	72 h	PR	R
19	59	ypT3c ypN1 M0	G3	no mut.	+	56	4	72 h	NR	PR
20	53	pT3c pN0 M0	G3	no mut	+	56	4	72 h	NR	PR
22	60	pT3c pN0 M0	G3	no mut.	+	56	4	72 h	NR	NR

All patients were female. # = specimen number, PS = patient serum, 2% PS was added to the culture media. CP = carboplatin, Ola = olaparib, R = response, PR = partial response, NR = non-response, n.d. = no data, no mut. = no somatic mutation, mut = mutated.

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
