# Peer review of "FOXM1 Inhibition in Ovarian Cancer Tissue Cultures Affects Individual Treatment Susceptibility Ex Vivo"

_cancers, 2021, doi:10.3390/cancers13050956_

Round 1

Reviewer 1 Report

Main reason for low scores is very small patient population with large variability across patient tissues cultured. Identified responses by authors are relevant for individual patients, but not the cross-section of patients included in this study.

  • Brief Summary of the research conducted and summarized in this manuscript.

The investigators have presented a rational objective to evaluate the response of ovarian tissue culture slices ex vivo to various therapeutic entities with an objective to suggest new treatment modalities for ovarian cancer.  More specifically, the investigators seek to establish through ex vivo cultures of ovarian cancer tissues sufficient justification to warrant combination therapy of carboplatin and FOXM1 inhibitor thiostrepton or combination therapy of Olaparib with thiostrepton.  To accomplish this objective, the authors collected ovarian cancer specimens from 14 patients, and established culture conditions that were acceptable in 10 patients for testing these drugs, and evaluated combination therapies in cultures obtained from 7 patients. Among the 7 patients, the responses were variable, leading to difficulty in establishing statistically consistent trends that can be summarized as common responses to comboination therapies examined in this report for ovarian cancer. This is in apparent contrast to their final sentence in Introduction, Here we present a stable and reproducible method that enables the investigation of tissue responses to optimize therapeutic options for ovarian carcinoma patients and thus avoiding unnecessary treatment.” 

Overall, this manuscript seems to fit in between the requirements specified by Cancers for a registered report stage 1 and registered report stage 2 on the topic of ovarian cancer. The current data collected provide sufficient preliminary evidence to proceed with additional collection of samples and generation of results for a Stage 2 manuscript submission.

  • Originality/Novelty: The investigators have taken a logical ex vivo culture approach to evaluate the efficacy of various monotherapies or combination therapies for ovarian carcinoma prior to subjecting patients to these combination regimens in a clinical trial. A significant challenge for these authors is that their working set of 7 patients contain a diverse range of responders to carboplatin, and a more uniform group of responders to olaparib, although there is not a statistically measurable response to olaparib among the endpoints evaluated in this report. The preparations for immunohistochemistry were nicely prepared.

  • Interest to the Readers: Certainly new approaches to treatment of ovarian cancer would be welcome additions to our body of knowledge for effective treatment of women’s health. Thiostrepton has been evaluated in a wide variety of disease animal models, suggesting a broader opportunity for affecting ovarian cancer outcomes than simply proliferation, apoptosis and the genes immediately relevant for proliferation. This broader scope of the effects of thiostrepton has not been evaluated in this particular ex vivo model system. 

  • Overall Merit: Low to moderate at this stage with insufficient numbers of patients distributed across categories of responder vs non-responder for a traditional Research Article. Good candidate for Registered Report using these preliminary results as supporting evidence of a refined research strategy.
  • The manuscript is generally well written, and organized according to the requested format for this journal. Placement of the conclusion after materials and methods seems quite different. The journal Cancers provides an avenue for registered reports that may allow these authors to refine this submission based on their early results provided here with an improved research strategy for stage 2.

BROAD COMMENTS

  • Significance:

It is recognized that human ovarian cancer tissue slices culture exhibit inherent variability across patients, and that particular attention to individual patient responses may encourage refinement of the experimental design in subsequent studies to obtain statistically meaningful conclusions. The graphical methods used to summarized results among the 7 patients hide the individual patient responses, leaving very few consistent responses across the 7 patients.  An alternative representation that links each patient control to her treatment response by a line, and subsequent analysis of the occurrence of decrease or increase may be more favorable.  Analysis of the trend lines would provide a better understanding of the causes of variability in Figures 1-4 that may or may not contain results from all 7 patients.

The authors suggest that a combination therapy containing thiostrepton can deliver improved outcomes compared to montherapies of thiostrepton, carboplatin or oliparib alone. There are too few patients with culture materials meeting the criteria for experimentation to reach the conclusions desired. The supplementary figures include results from individual patients, and the graphs and error bars represent the 3 slices obtained from an individual patient cultured in the same well. This indicates each response we observe in the graph is an individual response, not shared among all 7 patients. In graphs presented in Figures 1-4, there is only one instance of a statistically significant difference with carboplatin for an individual patient – not among the 7 patients.  Figure 3, no significant differences are indicated and the variability across patients is reflective of non-significant differences.  Figure 4, no significant differences indicated in Figure, though authors claim that thiostrepton alone significantly inhibits expression of the 4 genes.  The manuscript is unclear whether this is true for only one patient or all patients. The authors include description of statistical criteria at the bottom of each figure legend along with the symbol they are expected to insert into the figure to declare significant differences.  Absence of these symbols in the graph leaves reviewers the understanding that there are no statistically significant differences.

Scientific Soundness: This would serve as a strong submission of the Registered Report Stage 1, with preliminary results that could justify the study design for a Registered Report Stage 2.  The conclusions reached by these authors are weak because of the variability that underlies this preliminary analysis.  Encourage authors to consider Registered Report

SPECIFIC COMMENTS

  • Suggest minor edits with more traditional English phrases.

For example:

  1. Line 95
  2. Line 141
  3. Line 174
  • More background about thiostrepton in Introduction, particularly its effects in a diverse set of animal models
  • All of the graphs contain axis labels and print that is way too small for published material. Had to enlarge images to 175% just to read axis labels. 
  • Need to include asterisks indicating significant differences in the graphs if they are indeed significantly different.

Author Response

Please see the attachment. Thank you in advance.

Reviewer 2 Report

Article “FOXM1 inhibition in ovarian cancer tissue cultures affects individual treatment susceptibility ex vivo” by Brückner et al is an interesting study of to study a novel line of treatment in an ovarian cancer model. Overall, authors have presented interesting findings and have shown convincing evidence of their claims. A detailed critique is presented as follows-

  1. There are minute errors in paper and few misspellings that can be corrected.
  2. In the Abstract and Introduction, authors just mention that they study FOXM1. They should state clearly why it is being studied and how it regulates various pathways or in turn is regulated by other drugs used in the treatment of ovarian cancer.
  3. Discussion ends abruptly without any reference of the implications and future prospects of the study.   

Reviewer 3 Report

In this manuscript, the authors have described the development of a tissue culture model for ovarian cancer. Using this model, they examined each patient's response to standard treatment for ovarian cancer and evaluated the effect of the FOXM1 inhibitor thiostrepton. They then showed the additive effect of thiostrepton on carboplatin and olaparib. This is an interesting manuscript highlighting the usefulness of tissue culture as a tool for patient stratification. It might have been more informative if the authors had compared gene expression profiles (obtained by RNA sequencing) among the tissue culture models treated with thiostrepton, cisplatin, olaparib, and their combinations. This might have led to the discovery of a novel factor other than FOXM1, BRCA1/2, and RAD51. Before this paper is suitable for publication, please check if statistical significance was detected only in Fig2B #10 proliferating tumor fraction and Fig 5 #17 proliferating tumor fraction.

Round 2

Reviewer 1 Report

This reviewer recognizes the difficulties in conducting in vitro research with clinically obtained specimens, and recognizes the effort required to validate culture systems.  Our research also requires acquisition of human tissues and the conduct of in vitro drug testing, and the inherent variability of clinically obtained human specimens creates a significant challenge in generating reproducible results across multiple patients. Larger numbers of subjects are often required than can be accomplished with preclinical models.

As second challenge facing this investigation is the apparent heterogeneity in the response of established ovarian cancer cell lines in mouse xenografts treated with thiostrepton. This variability associated with thiotrepton treatments make homogeneous responses even more difficult in human cultured slices. 

The authors introduced more information in the Introduction to describe relevance of FOXM1 and thiostrepton.  Suggest new paragraph midway through line 61. Please rephrase line 63-64 to clarify: as written suggests that thiostrepton inhibits proteasome degradation of FOXM1, (thereby increases expression of FOXM1) which would be counter-intuitive to role of thiostrepton to inhibit and degrade FOXM1. A negative regulator of FOXM1 expression was proposed (2011, Gartel) that enables a potential mechanism involving FOXM1 transcription inhibition and proteasome inhibition by the same peptide entity.

 The authors have addressed background information regarding FOXM1 and